# Latitude-Induced and Behaviorally Thermoregulated Variations in Upper Thermal Tolerance of Two Anuran Species

**DOI:** 10.3390/biology11101506

**Published:** 2022-10-14

**Authors:** Ye Inn Kim, Ming-Feng Chuang, Amaël Borzée, Sera Kwon, Yikweon Jang

**Affiliations:** 1Korea Environment Institute, Sejong 30147, Republic of Korea; 2Department of Life Sciences, National Chung Hsing University, Taichung 40227, Taiwan; 3Laboratory of Animal Behaviour and Conservation, College of Biology and the Environment, Nanjing Forestry University, Nanjing 210037, China; 4Interdisciplinary Program of EcoCreative, Ewha Womans University, Seoul 03760, Republic of Korea

**Keywords:** amphibian, anura, ecophysiology, CTmax, environmental gradients, thermal response

## Abstract

**Simple Summary:**

Thermal tolerance can help understand species’ response to climate change, but regional scale data is missing. We filled this gap by investigating intraspecific variations in the critical thermal maximum (CTmax) of *Rana uenoi* tadpoles collected from rice paddies and *Bufo sachalinensis* tadpoles collected from lakes, over a five-degree latitudinal gradient. We found a positive relationship between latitude and CTmax in *R. uenoi* tadpoles, although they did not display specific thermoregulatory behaviors. In contrast, none of the abiotic factors were related to CTmax in *B. sachalinensis*, but the tadpoles swam closer to the water surface when water temperature increased. Thus, *R. uenoi* displays a physiological adaptive response while *B. sachalinensis* behaviorally thermoregulate. This is important as species specific responses to climate change have to be understood for their conservation.

**Abstract:**

Although thermal tolerance along geographical gradients gives an insight into species’ response to climate change, current studies on thermal tolerance are strongly skewed towards global-scale patterns. As a result, intraspecific variations are often assumed to be constant, despite a lack of evidence. To understand population-specific responses to thermal stress, we investigated the presence of intraspecific variations in the critical thermal maximum (CTmax) of tadpoles in two anuran species, *Rana uenoi* and *Bufo sachalinensis*. The study was conducted across a five-degree latitudinal gradient in the Republic of Korea. We exposed the tadpoles to increasing temperatures and recorded the CTmax for 270 *R. uenoi* individuals from 11 sites, collected in rice paddies, and for 240 *B. sachalinensis* individuals from ten sites, collected in reservoirs. We also recorded the swimming performance and behavior of the tadpoles when placed in an experimental apparatus during CTmax measurements. We then used linear regressions to determine the relationship between abiotic factors and CTmax. In *R. uenoi*, we found a positive relationship between latitude and CTmax, but the tadpoles did not display specific thermoregulatory behaviors. In *B. sachalinensis*, none of the abiotic factors such as climate and geographic coordinates were related to CTmax, but we detected a tendency to swim close to the water surface when water temperature was increasing. For *R. uenoi*, we tentatively relate the CTmax variability across the latitudinal gradient to a physiological adaptive response associated with habitat characteristics that are assumed to be fluctuating, as the species inhabits small water bodies prone to drying out. In the case of *B. sachalinensis*, the behavior observed may be linked to oxygen depletion and thermoregulation, as it may buffer temperature changes in the absence of physiological adjustment. These findings suggest that intra-specific variations in CTmax are greater than generally accounted for, and thermal conditions of natural environments are important for understanding thermal tolerance in ectothermic species. Our results highlight that species’ specific responses to climate warmings need to be studied to better protect species against climate change.

## 1. Introduction

Biological responses to the environment are determined by the organism’s ability to adapt through physiological and behavioral adjustments, such as phenotypic plasticity and thermoregulation [1,2]. The ability to adapt is an important factor that varies across species and populations, and it is linked to the potential of an organism to withstand extreme weather events. Thermal sensitivity is defined as the capacity of any organism to function under specific temperatures, but it generally includes measurements of locomotion, growth, development, fecundity and survivorship. A typical relationship between body temperature and performance in ectotherms is a skewed Gaussian distribution limited by lower and upper thermal tolerances (Figure 1 [3]). However, the curve is not fixed and can shift in response to various variables such as environment-induced epigenetic modification and associated gene expression [4]. These shifts are remarkable, as they enable individuals and populations to overcome climatic variations. Thermal performance curves are parameterized by thermal optimum temperature, thermal breadth and critical thermal limits. Of these parameters, the upper-bound value (critical thermal maximum; CTmax) is often used to predict the fate of species in response to rapid temperature increases under climate change scenarios [5].

CTmax is defined as the threshold temperature above which the locomotion of an animal becomes disorganized and the animal loses its ability to escape from lethal conditions [6]. CTmax measurements are used for numerous purposes [7], for instance to determine warming tolerances and evaluate species’ vulnerability to global warming (i.e., the difference between CTmax and the average temperature of the habitat; [8,9,10]), and to determine the distribution and persistence of species by incorporating tolerance limits into species distribution models [11,12,13]. Whereas research at the global scale compares CTmax values across species, the CTmax of a single population is often employed to represent that species under the assumption that interspecific CTmax variation is negligible [14].

Intraspecific variation in thermal tolerance is understudied, and intraspecific variations in CTmax across populations can be a key variable determining distribution and phenology [15]. In addition, recent studies conducted at narrow scales have revealed intraspecific variations in CTmax caused by habitat characteristics, providing results often contrary to those from global scale studies [16,17,18]. The aim of our study was therefore to measure the thermal tolerance of larvae in two anuran species with distinct habitat characteristics (shallow vs. deep) and to examine CTmax variations between populations of each species.

Compared to adult anurans that can lose heat through evaporation to thermoregulate, larvae are at a greater risk of becoming trapped in unhospitable aquatic environments. Thus, anuran larvae have evolved physiological metabolic compensation [19] and a higher thermal tolerance in comparison to adults [20]. In addition, and similarly to adults, anuran larvae behaviorally regulate their body temperature by moving to microhabitats with a different thermal environment [21,22]. For instance, Bufonid tadpoles track the most adequate temperature of the water body, avoiding deeper water when too cold [23] or avoiding direct sunlight when too hot [24].

Here, we investigated intraspecific geographic variation in the CTmax of tadpoles from two widespread and abundant species: *Rana uenoi* breeding in rice paddies and *Bufo sachalinensis* breeding in reservoirs. We hypothesized a difference in the magnitude of CTmax variation between the two species, but a similar variation in CTmax and latitude between them. In addition, as ephemeral water bodies are more likely to face extreme temperature changes, we expected the variation in CTmax in relation with latitude to be greater in *R. uenoi* than in *B. sachalinensis*, with behavioral adjustments in *R. uenoi* to compensate for high thermal variations. Specifically, we expected tadpoles thermoregulating to be at the surface of the experimental set-up, as tadpoles brought to CTmax are more active and try to move back to their preferred temperature range [25], in opposition to a resting state in which tadpoles stay close to the substrate to increase mimicry and decrease risks of predation [26]. Similarly, tadpoles under breathing stress, generally caused by higher temperatures in water bodies, swim closer to the surface [27], as the content of dissolved oxygen is higher.

## 2. Materials and Methods

### 2.1. Species Introduction

We selected two widespread species breeding in different types of habitats for this study. *Rana uenoi* (following [28]) preferentially spawns in lentic water bodies, but due to the anthropisation of habitats, the species regularly breeds in fallow rice paddies filled with temporary rainwater [29]. Populations that breed in rice paddies start spawning earlier in the season in the southwest than in the northeast of the country, with up to a month delay [30]. During development in rice paddies, tadpoles are likely to receive intense insolation due to the absence of surrounding vegetation and the low water depth. The other species, *Bufo sachalinensis* (following [31]) typically breeds in March in large water bodies such as reservoirs. Being explosive breeders, the breeding season of *B. sachalinensis* is generally synchronized across the country [32]. Furthermore, *B. sachalinensis* tadpoles in reservoirs are submitted to generally stable temperatures, as large water bodies act as a thermal buffer and receive lower insolation due to the vegetation on the banks, where tadpoles develop.

### 2.2. Tadpole Collection

We collected *R. uenoi* tadpoles between 18 March and 24 April 2018 at 11 sites across five latitudinal degrees throughout the Republic of Korea (Figure 2; Table 1; dataset available from Mendeley Data: [33]). The sampling sites were evenly distributed throughout the country, and the distance between the two nearest sites was 40 km. The elevation of sampling sites ranged between 33 and 389 m above sea level. The latitudinal gradients over which the study was conducted are considered an adequate representation of temperature variations to infer the species’ responses to temperature, despite the presence of potential confounding environmental factors [34,35,36]. In addition, sampling in the Republic of Korea along the same general latitudinal and climatic variation was able to demonstrate the Rapoport’s rule in ants [37], and passerine bird species richness was influenced by the latitudinal gradient [38]. Finally, the ecology of one of the focal species is known to be latitudinally dependent, as shown through ecological models [39]. At each site, three ponds were selected and eight tadpoles at Gosner development stages between 26 and 29 [40] were collected from each of the three ponds to avoid sampling tadpoles from the same clutch. Tadpoles were sampled from rice paddies where the water did not exceed 15 cm of depth to minimize habitat variation.

We collected *B. sachalinensis* tadpoles at development stages 26 to 29 between 25 March and 20 April 2018 from 10 sites (Figure 2; Table 1). The sampling sites for *B. sachalinensis* were restricted to reservoirs with water depth exceeding 1 m and non-deciduous vegetation along the banks. Toad tadpoles form schools generally composed of relatives, so we selected three independent schools and sampled eight tadpoles from each school to minimize sampling from the same clutch.

In addition to the GPS coordinates and altitude of each locality, we recorded daylight hours (h) at the sampling sites during tadpole collection, as this may affect the thermal environment of the locality. Daylight hours are correlated with season, making the variable thus correlated with environmental temperature. We extracted temperature and precipitation from the closest weather station of the Korean Meteorological Administration (http://www.kma.go.kr/weather/observation/aws_table_popup.jsp, accessed on 1 October 2022) every day for three weeks prior to each observation and averaged the data for each datapoint for further analyses.

We then transported all tadpoles to the laboratory within four hours of capture. All tadpoles were kept in clutch-specific containers (length: 20 cm, width: 12 cm, height: 15 cm, water depth: 12 cm) with constant aeration inside a portable cooler. All collected individuals were treated under the same laboratory conditions, and we used a photoperiod regime of 12 h light–12 h dark. To minimize possible physiological changes after sampling, tadpoles were kept in the water collected from the sampling site and not fed until the experiments were finished (72 h from sampling).

### 2.3. Estimation of Critical Thermal Maximum

The critical thermal maximum (CTmax) of individuals was measured within three days of collection to avoid acclimation to the laboratory setting [41]. The procedure used for the CTmax estimation followed the protocol developed by Wu et al. [42]. Generally, tadpoles were placed individually in 250 mL beakers with 200 mL of aged water, and beakers were arranged in a heated water bath (BW10G; JEIO Tech, Seoul, Korea). Up to eight individuals were tested at a time due to space restriction in the water bath. For the experiment, the initial water temperature was set at room temperature (17.14 ± 1.99 °C, mean ± SD). The heated water bath was set so that the water temperature in the beakers increased at a rate of 0.3 °C per minute. Aeration during temperature increase was constant to maintain adequate oxygen level and to mix water. Water temperature was monitored using a 450-AKT digital thermometer with a k-type thermocouple (Omega Engineering; Stanford, CA, USA). We used a blunt probe to assess the activity of the tadpole by gently touching the individual. The CTmax of each individual was defined as the water temperature at which an individual loses its righting response, including escape from the stimuli, ability of locomotion and keeping balance [43]. When the tadpole reached CTmax, the individual beaker containing the tadpole was immediately put in water at room temperature to cool slowly without temperature shock. None of the tadpole died during the experiments.

### 2.4. Behavioral Measurements

During the CTmax tests, we recorded the behavior of all individuals, except for two *R. uenoi* individuals that were obscured by an unidentified object on the video. We specifically focused on two behaviors: swimming (variable named movement) and vertical location of individuals in beakers (variable named location), as tadpoles change behavior in case of threats. We focused on these variables, as tadpoles use the water column differently in case of stress. Bufonid tadpoles swim close to the surface when avoiding predators (e.g., [44]) or in case of respiratory distress (e.g., [27]), and Ranid tadpoles generally sink to the bottom to increase crypticity (e.g., [26]). Similarly, when under heat stress, tadpoles display different swimming behaviors (e.g., [45]). The ‘movement’ of an individual tadpole was recorded as ‘0’ when not in motion and ‘1’ when swimming. The ‘location’ of an individual tadpole was recorded as ‘0’ when the individual was at the bottom of the beaker and ‘1’ when at the surface of the water (Table 1). Tadpoles were never found stationary at mid-height during testing. The behavior of *B. sachalinensis* was recorded seven times during the 60 min-long trials. The behavior of *R. uenoi* was recorded six times within the 50 min-long trials. We used a 10 min interval for both species, but the duration was different due to the difference in CTmax between the two species. The observations were made about 50 cm away from the experimental setup to minimize the effect of human observers.

### 2.5. Statistical Analysis

We performed linear regressions using linear mixed effect models (hereafter LMM) to determine which abiotic factors were important for CTmax using the ‘nlme’ CRAN package [46]. The response variable was CTmax, and fixed factors were latitude (°N), altitude (m), daylight hours (h), average temperature (°C) and precipitation (mm). *p* values for fixed factors were obtained by using the ‘lmerTest’ package [47]. Sites and the collection date, converted to the Julian calendar, were included as random factors. As random factors in LMM require observations within the random variables groups not to be correlated, we used the ‘MuMln’ package (version 1.47.1 [48]) to check whether the model with random factors or the model without random factors was a better fit. Normality and homogeneity of residuals were inspected visually through quantile–quantile plots (hereafter Q-Q plot) and fitted values versus residuals graphs. Multicollinearity was assessed by correlation coefficient and variance inflation factor (VIF, hereafter). Correlation coefficients that were lower than 0.6 and VIF lower than 3 were selected and entered as explanatory variables in the models (Table 2).

The change in movement over time was analyzed with logistic regressions using generalized linear mixed effect models (hereafter GLMM) using the ‘lme4′ package with time point as the fixed factor and movement as the response variable. The location change over time was analyzed with logistic regressions using GLMMs, using time point as the fixed factor and location as the response variable. Individual IDs and sites were included as random factors in each model following the assumption that observations within the random factors were correlated. Normality of error, homogeneity of variance and linearity were confirmed using diagnostic plots (fitted vs. residual, scale-location, Q-Q plot, influence). Separate analyses were performed for *R. uenoi* and *B. sachalinensis* for each model. A significant criterion was set as alpha = 0.05; values were presented as mean ± SD, and all analyses were conducted in *R* version 3.5.1 [49].

## 3. Results

### 3.1. Effect of Abiotic Factors on Critical Thermal Maximum

We determined linear CTmax gradient along the five degrees of latitude of our study for both species (Figure 3). The slopes and patterns were, however, drastically different for the two species. The mean CTmax of *Rana uenoi* was 36.76 ± 0.74 °C (mean ± SD) and ranged from 35.0 to 38.4 °C, whereas the mean CTmax of *Bufo sachalinensis* was 38.3 ± 0.28 °C and ranged from 37.5 to 38.9 °C. The average CTmax, sunset time, temperature, precipitation and the GPS coordinates of each site for the two species are provided in Table 1 and non-averaged values are in the Mendeley Data [33].

In *R. uenoi*, latitude was the only significant factor for CTmax (LMM: *p* < 0.001; Table 2), whereas altitude, temperature, precipitation and daylight hours were not significant (*p* > 0.05). The CTmax values were higher for the northern than the southern populations (Figure 3A). The collection date and sites were set as random factors and explained 36% and 16% of the total CTmax variance, respectively.

For *B. sachalinensis*, none of the abiotic factors had a significant effect on CTmax (LMM: *p* > 0.05; Table 2), and the CTmax values were not significantly different along the latitudinal gradient (Figure 3B). The collection date and sites equally explained 25% of the total variance in CTmax in *B. sachalinensis*

### 3.2. Temperature-Induced Behavior Changes in the Two Species

For *R. uenoi* (*n* = 268), the pattern of movement was generally consistent, with a peak at the beginning of the experiment, a drop after 10 min, followed by an increased and a plateau phase throughout the rest of the experiment. An average of 18.53 ± 5.32% of the individuals were in movement during the whole course of the experiment. A total of 22.76% of the individuals were in movement at the first ten-minute mark (time = 0 min), and this percentage slightly decreased before increasing again at the fifth ten-minute mark (time = 40 min), resulting in a non-significant change in movement over time (Table 3; Figure 4A). The proportion of *R. uenoi* tadpoles located at the surface generally increased throughout the experiment, with an 8.46 ± 2.98% average, ranging from 5.22 to 13.43%. Only 5.22% of tadpoles were at the surface at the beginning of the experiment, but the number significantly changed and reached 13.43% by the fifth ten-minute mark (GLMM; *p* < 0.001; Table 4 and Figure 4B).

Changes in both movement and location in *B. sachalinensis* over time were significant (GLMM; *p* < 0.001, Table 2 and Table 4). The percentage of individuals in movement decreased throughout the experiment, with an average 48.75 ± 15.99% of the individuals in movement during the experiment. A maximum of 66.67% of the tadpoles were in movement at the first ten-minute mark, and the number decreased gradually until reaching 27.92% at the sixth ten-minute mark (Figure 4A). In general, the proportion of tadpoles at the surface increased with time, with an average of 10.77 ± 6.37% at the surface. At the beginning of the experiment, 4.58% of tadpoles were at the surface, and the number increased gradually over time, reaching 20.42% of tadpoles located at the surface of the water at the last ten-minute mark (Figure 4B).

## 4. Discussion

We had predicted a weak intraspecific correlation between CTmax and latitude over the five degrees of latitude of our study, and our hypothesis was supported in *Rana uenoi* but not in *Bufo sachalinensis*. The CTmax variation observed in the two species was assumed to be more relevant to the species’ thermal environments. We also hypothesized that both species would be able to mitigate heat stress through behavioral adjustments, and this was partially correct, as the two species expressed distinct behavioral responses to thermal stress.

The first main finding of this study is the positive relationship between latitude and CTmax in *R. uenoi*. This result is, however, in contrast to the general patterns, whereby species are expected to have a higher thermal tolerance closer to the equator [50,51]. Due to a scarcity of experimental evidence in the literature, our counterintuitive results for *R. uenoi* can be interpreted within two broad contexts: in response to daily thermal fluctuations (DTF; [52]) or as a trade-off between two thermal traits in response to acute thermal stress [5].

DTF is the temperature change within an entire day, a particularly challenging variable for ectotherms, as a high DTF increases metabolic demands compared to constant temperature conditions [53,54]. Accordingly, higher DTFs result in increased CTmax in tadpoles of *Limnodynastes peronei*, *L. tasmaniensis* and *Platyplectrum ornatum* [52]. Our study emphasized that the CTmax of anuran larvae is more strongly influenced by temperature variability than by the mean ambient temperature. Similarly, another study [54] also found metabolic responses to DTF. For example, *Limnodynastes* species have the ability to buffer increases in metabolism resulting from a high DTF so that growth and development are not impaired [52].

Water temperature variability in rice paddies is comparatively high, as water is very shallow (<15 cm deep). Consequently, if the risk of exposure to temperature fluctuations is high and *R. uenoi* larvae cannot physiologically regulate the high metabolic demand resulting from a high DTF, then growth and development of individuals inhabiting rice paddies is likely to be negatively impacted. A negative impact of high DTFs would cause increased vulnerability to predators [55,56] and reduced fecundity at maturity. In view of the literature about the ecology of *R. uenoi* [29,57], we surmise that the species has adequately adapted to high DTFs. To test this possibility, we tentatively suggest the presence of a positive relationship between the different magnitude of DTF generated by the latitudinal gradient and their impact on CTmax variation. Thus, quantitative measurements of DTF across different latitudinal sites are further needed to demonstrate and prove this possibility.

In addition, the results in *R. uenoi* support the idea of a trade-off between two thermal traits associated with the production of heat shock proteins (HSP hereafter) in response to acute thermal stress (e.g., [58]). HSPs protect the organism during heat exposure, and the two thermal-dependent traits are HSP-dependent and HSP-repression. For instance, acute exposure (1–2 h) to high temperatures increases subsequent heat tolerance by inducing the production of HSP [58]. Here, it is possible that the higher CTmax of *R. uenoi* observed at northern latitudes results from the production of HSP in response to acute thermal stress, whereas populations at southern latitudes display comparatively low CTmax because of the absence of an HSP effect. A continuous HSP production induced by thermal stress can cause negative impacts on the cells of an organism [59]; we assumed that the southern populations would maintain a low level of HSP to avoid a continuous HSP production as they have more chances to encounter extreme high temperatures (Figure 1). As our expectation was wrong, individuals in southern areas may have developed alternative mechanisms than HSP production to protect themselves from extreme thermal events. As a result, climate change may not impact the survival of the species through this specific physiological process, but through other processes as the breeding phenology of the species became delayed in recent years, likely due to climate change [30].

In contrast with the clear CTmax gradient along latitude in *R. uenoi*, the CTmax pattern in *B. sachalinensis* indicates a latitudinally constant CTmax within the five degrees of our study. This result is considerably more in line with the literature available, and environmentally induced changes result in the plasticity of thermal tolerance [60]. However, the magnitude of plasticity in thermal tolerance in response to environmental signals is expected to be different between species, depending on their thermal sensitivity. Considering that *B. sachalinensis* is an ecological generalist and has a broad distribution range [31], especially when compared to *R. uenoi* [28], the environmental variability within the five-degree latitudinal gradient studied is not sufficiently large to show plasticity in CTmax, and therefore may preclude the species from adapting to climate change, despite demonstrated variations in responses to environmental variables within the range of the species [39]. Alternatively, it is possible that *B. sachalinensis* does not need to show plasticity in CTmax due to their generalist lifestyle or because of other mechanisms of regulation.

As we detected significant behavioral changes in *B. sachalinensis* only, our results are likely to reflect behavioral thermoregulation in a species that has developed displacements in the water bodies to maintain a specific thermal profile. For instance, tadpoles can move deeper in the water column or under vegetation to lower their body temperature [61]. The absence of behavioral adjustment in *R. uenoi* corresponds to the ecology of the species in the wild, as its environment is made of shallow water bodies. Instead, the physiological adjustment might compensate for the absence of thermoregulatory behavior in the species. Tadpoles of both *R. uenoi* and *B. sachalinensis* tended to stay on the surface more frequently during the increase in water temperature. Swimming up to the surface of the water is a common behavior of tadpoles (i.e., bobbing; [62]), and we suggested that tadpoles moved to the surface for a better uptake of dissolved oxygen, as warm water holds less dissolved oxygen.

In this study we provide empirical evidence in support of the relation between the thermal conditions of natural environments and thermal tolerance of ectothermic species along environmental gradients [63]. Species at lower latitudes have higher CTmax, despite the difference between ambient temperature and maximal temperature being lower, and thus making the low-latitude population more threatened by climate change [8]. Partially along our predictions, *R. uenoi* across latitudes likely undergo physiological adaptive responses linked to variability in CTmax, in association with fluctuations in the thermal environment of rice paddies. On the other hand, the behavioral thermoregulation observed in *B. sachalinensis* mediates buffer temperature changes in the absence of physiological adjustment. We presented two possible scenarios for the opposite results to the previously established latitudinal pattern (Figure 1). The average temperature has risen by 1.5 °C in the Republic of Korea over the last 100 years because of the effect of climate change [64], and our two focal species have shown a likely related delay in breeding phenology ([30] and unpublished data for *B. sachalinensis*). Though numerous studies are actively engaged in explaining and documenting the current status of biodiversity and phenological change to assess the impact of climate change [65,66,67,68], little attention has been paid to the ability of species to survive dramatic changes in temperature. To answer the question regarding how species respond to global warming, acquiring CTmax estimates across populations for species independently is required. In agreement with Sears et al. [69], quantifying the magnitude of intraspecific variability in CTmax ought to improve predictions of species’ responses to climate change. Finally, additional research about the use of thermal refuges, whether there is variation in thermal refuges across latitudinal gradients and how different life stages will be affected by climate change, in synergy with genetic variability and adaptive variation, would provide a new avenue of conservation research, and could lean on already existing frameworks [70,71].

## 5. Conclusions

As most studies on climate change are global, missing intraspecific variations, we studied geographic variations in the thermal resistance of two amphibian species. Brown frog tadpoles showed a latitudinally variable thermal resistance, but did not vary in behavior, a likely physiological adaptive response. Toad tadpoles showed the opposite pattern, likely related to oxygen depletion and thermoregulation. Intraspecific variations in thermal resistance are therefore greater than generally expected.

## Figures and Tables

**Figure 1 biology-11-01506-f001:**
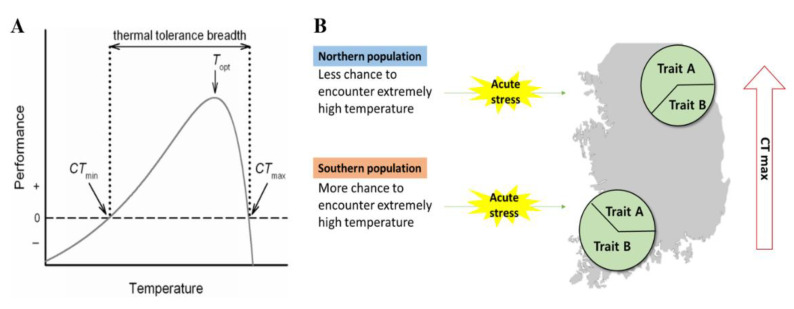
Relationship between temperature and performance in ectotherms and heat shock protein-dependent thermal responses. (**A**) Typical relationship between body temperature and performance in ectotherms is a left-skewed Gaussian distribution limited by lower thermal tolerance and upper thermal tolerance. (**B**) Trait A: heat shock protein-dependent thermal response, trait B: heat shock protein-repression thermal response.

**Figure 2 biology-11-01506-f002:**
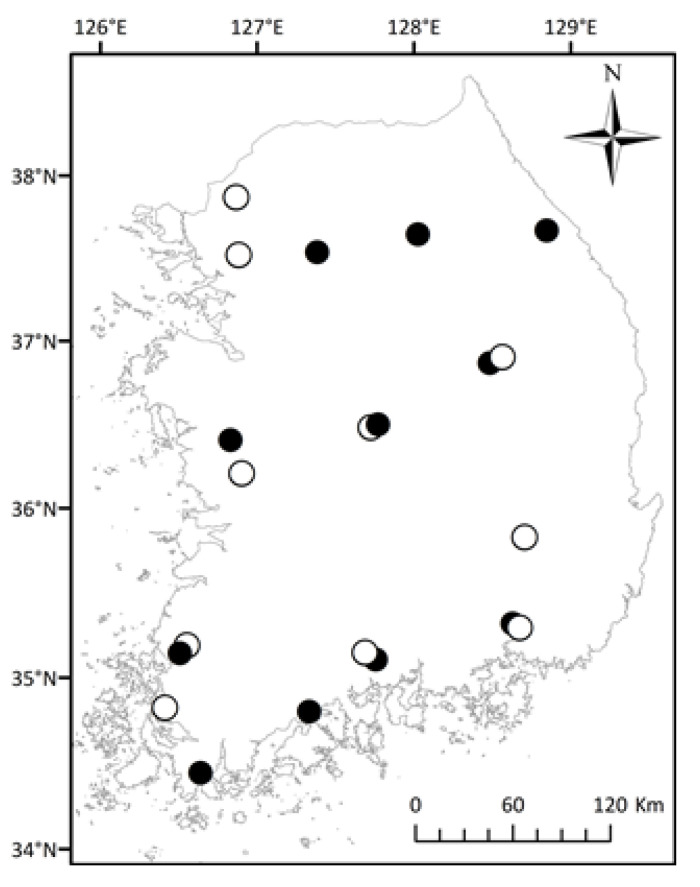
Sampling sites for *Rana uenoi* (11 sites, close circle) and *Bufo sachalinensis* (10 sites, open circle). Sampling was conducted between March and April 2018.

**Figure 3 biology-11-01506-f003:**
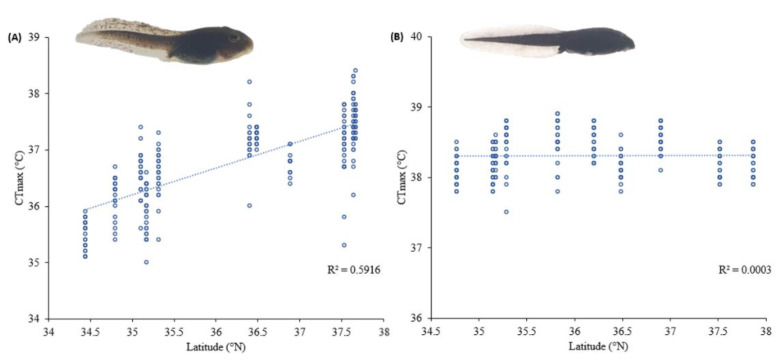
The relationship between latitude and critical thermal maximum (CTmax) in *Rana uenoi* ((**A**); *n* = 270) and *Bufo sachalinensis* ((**B**); *n* = 240). There was a significant positive relationship between latitude and tadpoles’ CTmax in *R. uenoi*.

**Figure 4 biology-11-01506-f004:**
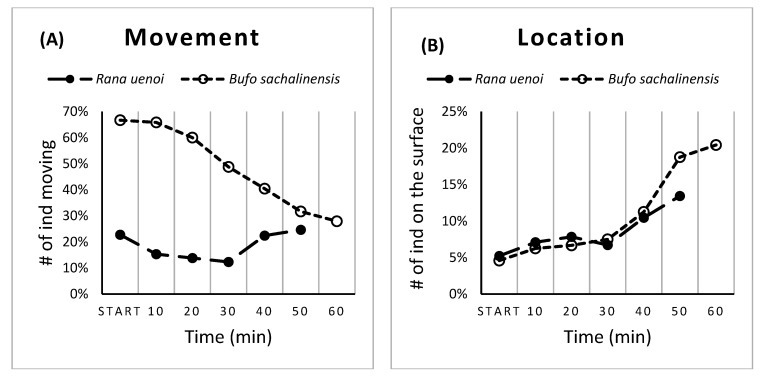
The frequency changes of movement (**A**) and location (**B**) of *Rana uenoi* (close circle; *n* = 270) and *Bufo sachalinensis* (open circle; *n* = 240) over time.

**Table 1 biology-11-01506-t001:** The detailed sample size and CTmax of *Rana uenoi* (*n* = 270) and *Bufo sachalinensis* (*n* = 240) and associated sample site information (coordinates, altitude, daily average temperature (°C), daily precipitation (mm) and daylight hours (h)) during tadpole collection. The values of CTmax are presented as mean ± SD. Complete dataset available from Mendeley Data: [33]).

Species	Site ID	Sample Size	Latitude	Longitude	Alt (m)	Temperature (°C)	Precipitation (mm)	DaylightHours (h)	CTmax (°C)
*Rana uenoi*	RU-1	30	37.53342	127.3807414	100	10.35	97.25	6.47	37.0 ± 0.5
	RU-2	29	37.64184	128.0233030	289	9.4	93.3	6.96	37.4 ± 0.4
	RU-3	24	37.66629	128.8459008	190	10.8	112.05	7.02	37.5 ± 0.2
	RU-4	20	36.40426	126.8289935	33	8.2	127.9	6.55	37.1 ± 0.4
	RU-5	24	36.48659	127.7382435	216	9.8	116.35	6.61	37.2 ± 0.1
	RU-6	20	36.90107	128.5241611	267	10.05	142.55	7.38	36.7 ± 0.1
	RU-7	28	35.17369	126.5375166	134	10.2	119.65	7.28	35.9 ± 0.4
	RU-8	24	35.10432	127.7560522	137	12.05	139.35	6.98	36.5 ± 0.3
	RU-9	24	35.31685	128.6274404	40	11.95	154.25	7.09	36.6 ± 0.4
	RU-10	23	34.44065	126.6373606	77	10.5	133.85	6.49	35.4 ± 0.2
	RU-11	24	34.79935	127.3308623	62	11	162.35	6.79	36.1 ± 0.3
*Bufo sachalinensis*	BG-1	24	37.86783	126.8645720	88	8.2	79.35	7.03	38.1 ± 0.2
	BG-2	24	37.51784	126.8813251	13	10.55	89.9	7.09	38.0 ± 0.1
	BG-3	24	36.90107	128.5241611	344	10.8	112.05	7.02	38.5 ± 0.1
	BG-4	24	36.20582	126.9009314	44	10.2	127.9	6.55	38.5 ± 0.2
	BG-5	24	36.48607	127.7377301	216	9.8	116.35	6.61	38.1 ± 0.1
	BG-6	24	35.82585	128.7048109	81	10.05	142.55	7.38	38.5 ± 0.2
	BG-7	24	35.18138	126.5409766	157	10.2	119.65	7.28	38.2 ± 0.1
	BG-8	24	35.14597	127.6870155	23	12.05	139.35	6.98	38.1 ± 0.2
	BG-9	24	35.29143	128.6729614	44	11.95	154.25	7.09	38.4 ± 0.3
	BG-10	24	34.76735	126.4103310	188	8.2	160.85	6.81	38.0 ± 0.1

**Table 2 biology-11-01506-t002:** Summary of linear mixed effect models for the CTmax of tadpoles. For *Rana uenoi*, we tested a total of 270 tadpoles from 11 localities (random factor), with marginal *R*^2^/conditional *R*^2^ = 0.59/0.80. For *Bufo sachalinensis*, we used a total of 240 observations from 10 localities (random factor), with marginal *R*^2^/conditional *R*^2^ = 0.09/0.50. CI = confidence interval; VIF = variance inflation faction; significant results are labeled with “*”.

Species	Fixed Factors	Estimate ± SE	DF	*t*-Value	*p*-Value	VIF
*Rana uenoi*	(Intercept)	15.93 ± 7.35	259	2.17	0.0312	
	Latitude	0.59 ± 0.17	5	3.46	0.018 *	2.65
	Altitude	0 ± 0	5	−0.03	0.979	2.4
	Sunset	−0.24 ± 0.55	5	−0.44	0.680	1.88
	Average temperature	0.03 ± 0.14	5	0.22	0.832	1.43
	Precipitation	0.01 ± 0.01	5	0.75	0.407	2.8
*Bufo sachalinensis*	(Intercept)	35.19 ± 6.62	227	5.32	<0.001	
	Latitude	0.09 ± 0.16	7	0.53	0.474	4.95
	Altitude	0 ± 0	7	0.51	0.687	1.08
	Sunset	−0.15 ± 0.21	7	−0.75	0.273	1.04
	Average temperature	0.07 ± 0.06	7	1.05	0.251	1.15
	Precipitation	0 ± 0.01	7	0.44	0.66	4.95

**Table 3 biology-11-01506-t003:** Results of the binomial logistic regression assessing the relation between the movement (swim vs. immobile) of tadpoles and time course for *Rana uenoi* (*n* = 268) and *Bufo sachalinensis* (*n* = 240). B = beta coefficient; SE = standard error; significant results are labeled with “*”.

Species	Fixed Factors	B	SE	*z* Value	*p* Value
*Rana uenoi*	Intercept	−1.73	0.14	−11.5	<0.001 *
	Time	0.06	0.03	1.59	0.11
*Bufo sachalinensis*	Intercept	0.96	0.2	4.43	<0.001 *
	Time	−0.34	0.02	−11.86	<0.001 *

**Table 4 biology-11-01506-t004:** Results of the binomial logistic regression assessing the relation between the location (bottom/surface) of tadpoles and time course for *Rana uenoi* (*n* = 268) and *Bufo sachalinensis* (*n* = 240). B = beta coefficient; SE = standard error; significant results are labeled with “*”.

Species	Fixed Factors	B	SE	*z* Value	*p* Value
*Rana uenoi*	Intercept	−3.28	0.25	−13.11	<0.001 *
	Time	0.2	0.05	3.63	<0.001 *
*Bufo sachalinensis*	Intercept	−4	0.39	−10.18	<0.001 *
	Time	0.37	0.04	8.05	<0.001 *

## Data Availability

Data available as from Mendeley Data repository http://dx.doi.org/10.17632/9ffxngmhtb.1, accessed on 12 October 2022 [27]).

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
