# Peer review of "Latitude-Induced and Behaviorally Thermoregulated Variations in Upper Thermal Tolerance of Two Anuran Species"

_biology, 2022, doi:10.3390/biology11101506_

Round 1

Reviewer 1 Report (Previous Reviewer 2)

The authors of the study investigated the intraspecific variation in thermal tolerance of two anuran species, Rana uenoi and Bufo sachalinensis, along a latitudinal gradient in South Korea. They measured the Critical Thermal Maximum (CTmax) and the behavioral responses to warming temperature from the tadpoles of the two species. Their findings showed latitudinal variation in CTmax for R. uenoi, but no variation for B. sachalinensis. These differences between species are most likely due to variations in their thermal niches and their adaptation to different microhabitats. Understanding inter- and intraspecific variation in CTmax across a latitudinal gradient help to better understand how species will be able to tolerate and adapt to climate change.

The authors revised the manuscript based on the reviewer's comments from their previous submission. They addressed the behavior comments from my previous suggestion and successfully addressed some of my previous comments. This paper would be more substantial, however, if the authors measured the microclimatic temperature variation in their habitat across the gradient. Throughout the paper, they mention the thermal conditions of the natural environment, but they only have the daylight and cloud cover data for the time when they collected the tadpoles. With microhabitat data, they could better link the physiological responses to the experienced environmental temperatures. Since they don’t have this dataset, they could have at least estimated the microclimatic data using this resource below:

Kearney, M. R., Isaac, A. P. & Porter, W. P. (2014). microclim: Global estimates of hourly microclimate based on long-term monthly climate averages. SCIENTIFIC DATA, 1 (1), https://doi.org/10.1038/sdata.2014.6.

Major comments:

  1. What are the environmental temperature differences across the gradient?  How does this temperature variation reflect on the microclimate for each species? 

The environmental data they measured in each location were humidity and daylight hours only during tadpole collection. Despite the correlation between season temperature variation with daylight, and cloud cover with humidity, this dataset does not represent the microclimate temperatures that the two species of frogs experience. Without data to translate the regional macroclimate along the Latitudinal gradient to local microclimate temperatures that the tadpoles experience, it is impossible to predict how these organisms will respond to climate change. ​I suggested describing better how the temperature changes across the gradient, but the authors only described that this gradient “is considered an adequate representation of temperature variations to infer the species’ responses to temperature.” They could better explore the magnitude of macro- and microhabitat temperature variation along the gradient.

  1. How can you use the results found in this study to extrapolate conservation efforts to other amphibians with similar niches? What other aspects of the studied amphibians need to be examined to make better predictions about the response to climate change?

In the discussion, the authors could extrapolate their findings to discuss how similar conservation efforts could be applied to other amphibians. They could also discuss how a 1.5-degree increase in temperature would affect the microhabitat of each species and whether each species would be affected differently. To make better predictions about how climate change will affect amphibians, other factors that need to be considered are: how amphibians use thermal refuges, whether there is variation in thermal refuges across a latitudinal gradient, and how different life stages will be affected. Genetic variability and adaptive variation are also important. They could have used the framework provided by Willian et al (2008). 

Williams, Stephen E., et al. "Towards an integrated framework for assessing the vulnerability of species to climate change." PLoS biology 6.12 (2008): e325.

Author Response

Reviewer 1:

The authors of the study investigated the intraspecific variation in thermal tolerance of two anuran species, Rana uenoi and Bufo sachalinensis, along a latitudinal gradient in South Korea. They measured the Critical Thermal Maximum (CTmax) and the behavioral responses to warming temperature from the tadpoles of the two species. Their findings showed latitudinal variation in CTmax for R. uenoi, but no variation for B. sachalinensis. These differences between species are most likely due to variations in their thermal niches and their adaptation to different microhabitats. Understanding inter- and intraspecific variation in CTmax across a latitudinal gradient help to better understand how species will be able to tolerate and adapt to climate change.

The authors revised the manuscript based on the reviewer's comments from their previous submission. They addressed the behavior comments from my previous suggestion and successfully addressed some of my previous comments. This paper would be more substantial, however, if the authors measured the microclimatic temperature variation in their habitat across the gradient. Throughout the paper, they mention the thermal conditions of the natural environment, but they only have the daylight and cloud cover data for the time when they collected the tadpoles. With microhabitat data, they could better link the physiological responses to the experienced environmental temperatures. Since they don’t have this dataset, they could have at least estimated the microclimatic data using this resource below:

Kearney, M. R., Isaac, A. P. & Porter, W. P. (2014). microclim: Global estimates of hourly microclimate based on long-term monthly climate averages. SCIENTIFIC DATA, 1 (1), https://doi.org/10.1038/sdata.2014.6.

è Thank you for revising our manuscript a second time, we have addressed and answered all comments in detail.

Major comments:

  1. What are the environmental temperature differences across the gradient?  How does this temperature variation reflect on the microclimate for each species? 

The environmental data they measured in each location were humidity and daylight hours only during tadpole collection. Despite the correlation between season temperature variation with daylight, and cloud cover with humidity, this dataset does not represent the microclimate temperatures that the two species of frogs experience. Without data to translate the regional macroclimate along the Latitudinal gradient to local microclimate temperatures that the tadpoles experience, it is impossible to predict how these organisms will respond to climate change. ​I suggested describing better how the temperature changes across the gradient, but the authors only described that this gradient “is considered an adequate representation of temperature variations to infer the species’ responses to temperature.” They could better explore the magnitude of macro- and microhabitat temperature variation along the gradient.

è We agree with the point raised by the reviewer, and we have managed to access data to answer the point raised. The addition in the materials and methods is such as: “In addition to the GPS coordinates and altitude of each locality, we recorded daylight hours (h) at the sampling sites during tadpole collection as it may affect the thermal environment of the locality. Daylight hour is correlated with season, making the variable thus correlated with environmental temperature. We extracted temperature and precipitation from the closest weather station of the Korean Meteorological Administration (http://www.kma.go.kr/weather/observation/aws_table_popup.jsp) for every day for three weeks prior to each observation and averaged the data for each datapoint for further analyses”.

We have also added the data in the Table 1.

The corrected analyses are such as: “We performed linear regressions using Linear Mixed Effect Models (hereafter LMM) to determine which abiotic factors were important for CTmax using the ‘nlme’ CRAN package [46]. The response variable was CTmax, and fixed factors were latitude (ËšN), altitude (m), daylight hours (h), average temperature () and precipitation (mm)”.

                None of the new variables was significant in the model, and we did not have to add significant text in the results and the discussion – we only made minor changes throughout to reflect the integration of the new variables, please see the new version of the ms. In addition, we have referred to the reference provided above to facilitate the integration of the data – please see comment below.

  1. How can you use the results found in this study to extrapolate conservation efforts to other amphibians with similar niches? What other aspects of the studied amphibians need to be examined to make better predictions about the response to climate change?

 In the discussion, the authors could extrapolate their findings to discuss how similar conservation efforts could be applied to other amphibians. They could also discuss how a 1.5-degree increase in temperature would affect the microhabitat of each species and whether each species would be affected differently. To make better predictions about how climate change will affect amphibians, other factors that need to be considered are: how amphibians use thermal refuges, whether there is variation in thermal refuges across a latitudinal gradient, and how different life stages will be affected. Genetic variability and adaptive variation are also important. They could have used the framework provided by Willian et al (2008). 

Williams, Stephen E., et al. "Towards an integrated framework for assessing the vulnerability of species to climate change." PLoS biology 6.12 (2008): e325.

è The analysis suggested by the reviewer is interesting, and it would be a very exciting follow-up study, which is worth a whole manuscript in its own. To answer the points, we have highlighted the suggestion raised here in the abstract such as: “Our results highlight that species specific response to climate warming are needed to better protect the species against climate change“; and in the discussion, so that readers can know about this option, and as the data is open access, it can be re-analysed by anyone. The new paragraph in the discussion is such as: “Finally, additional research about the use of thermal refuges, whether there is variation in thermal refuges across latitudinal gradients, and how different life stages will be affected by climate change, in synergy with genetic variability and adaptive variation, would pro-vide a new avenue of conservation research, and could lean on a framework already existing [71-72].”.

Reviewer 2 Report (Previous Reviewer 1)

Dear authors,

The manuscript is significantly improved.  There are only some minor editing things that need to be addressed.  

Lines 207-210 change "swimming" to "swim"; "sinking" to "sink"; "increasing" to "increase "; "tadpole" to "tadpoles"

Lines 320, 406 change "evidences" to "evidence"

Author Response

Reviewer 2:

Dear authors,

 The manuscript is significantly improved.  There are only some minor editing things that need to be addressed.  

è Thank you for reviewing our manuscript a second time, we have answered the comments below

Lines 207-210 change "swimming" to "swim"; "sinking" to "sink"; "increasing" to "increase "; "tadpole" to "tadpoles"

è Corrected as suggested

Lines 320, 406 change "evidences" to "evidence"

è Corrected as suggested

Reviewer 3 Report (New Reviewer)

The authors present an interesting study on thermoregulated variations and thermal- adaptations of two anuran species in Republic of Korea. Study design and research questions are clearly described. In this sense, it is easy to understand the aim of this study. The bright side of the manuscript is that to provide and practical details on method to study such species in this content, and thermal tolerance of two anuran species. In this context, the study contributes to different fields. However, some minor concerns were raised. Therefore, I would like to make some suggestions to improve the quality of the paper as below:

There are many sentences that were written in red font in the manuscript (e.g. lines 15-17, line 18, line 310, lines 418-419 etc.) and also there are many comments. These sentences may be from the previous peer review process, but authors did not mention why some sentences were written in red. Comments should be deleted.

Figure 1 and Table 1 written as Figure S1 and Table S1. These Figure and Table are not supplementary files so “S” should be deleted, otherwise they should be added to supplementary file section. Moreover, all Figure and Table numbers should be controlled and corrected.

Section 6 was given as “Patents”. Is there any patent related to study?  

Abstract

Line 28: is likely -> may be

Line 31: Please add a sentence that defines your contribution to related fields since species specific responses to climate change is important for conservation of species.

Introduction

Line 98: “species studied at the same scale” Please clarify which species.

Materials and Methods

Line 125: Please check the Figure and Table numbers.

Line 140: “dataset available from Mendeley Data: [33]” I could not access the dataset since it is in embargo till 8 October 2022, 12:00. Is this the same dataset which was given as supplementary file? Please clarify.

Lines 203-204: “During the CTmax tests, we recorded the behavior of all individuals, except for two R. uenoi individuals.” Please explain why two individuals were not recorded.

Discussion

Line 318: in opposition -> in contrast to

Line 327: another study -> Kern et al. 2015 [54]

Author Response

Reviewer 3:

The authors present an interesting study on thermoregulated variations and thermal- adaptations of two anuran species in Republic of Korea. Study design and research questions are clearly described. In this sense, it is easy to understand the aim of this study. The bright side of the manuscript is that to provide and practical details on method to study such species in this content, and thermal tolerance of two anuran species. In this context, the study contributes to different fields. However, some minor concerns were raised. Therefore, I would like to make some suggestions to improve the quality of the paper as below:

è Thank you for the time spent on our manuscript, we have addressed and answered all comments in detail.

There are many sentences that were written in red font in the manuscript (e.g. lines 15-17, line 18, line 310, lines 418-419 etc.) and also there are many comments. These sentences may be from the previous peer review process, but authors did not mention why some sentences were written in red. Comments should be deleted.

è This was indeed from the peer review process, resubmitted together with the answers to the reviewers. This is now fixed, and the answer to the points you have raised are below.

Figure 1 and Table 1 written as Figure S1 and Table S1. These Figure and Table are not supplementary files so “S” should be deleted, otherwise they should be added to supplementary file section. Moreover, all Figure and Table numbers should be controlled and corrected.

è This was an editorial edit, we have now harmonised numbers for both tables and figures.

Section 6 was given as “Patents”. Is there any patent related to study?  

è Corrected as acknowledgements

Abstract

Line 28: is likely -> may be

è Corrected as suggested

Line 31: Please add a sentence that defines your contribution to related fields since species specific responses to climate change is important for conservation of species.

è We have added a sentence such as: “Our results highlight that species specific response to climate warming are needed to better protect the species against climate change”.

Introduction

Line 98: “species studied at the same scale” Please clarify which species.

è This was a mistake and we have corrected it such as: “We hypothesized a difference in the magnitude of CTmax variation between the two species, but a similar variation in CTmax and latitude between the two species”.

Materials and Methods

Line 125: Please check the Figure and Table numbers.

è This was an editorial edit, we have now harmonised numbers for both tables and figures.

Line 140: “dataset available from Mendeley Data: [33]” I could not access the dataset since it is in embargo till 8 October 2022, 12:00. Is this the same dataset which was given as supplementary file? Please clarify.

è The Mendeley data provides additional information as the Table 1 provides averages only. We have clarified this point such as: “The average CTmax, sunset time, relative humidity, and the GPS coordinates of each site for the two species are provided in Table S1 and non-averaged values are in the Mendeley Data [33]”.

Lines 203-204: “During the CTmax tests, we recorded the behavior of all individuals, except for two R. uenoi individuals.” Please explain why two individuals were not recorded.

è We have clarified this point of obscuration: “During the CTmax tests, we recorded the behavior of all individuals, except for two R. uenoi individuals which were obscured by an unidentified object on the video”.

Discussion

Line 318: in opposition -> in contrast to

è Corrected as suggested

Line 327: another study -> Kern et al. 2015 [54]

è Thank you, we have corrected this reference

Round 2

Reviewer 1 Report (Previous Reviewer 2)

The paper has been significantly improved with corrections and additions in response to my comments. Including new environmental data and analyses has strengthened the paper, and the overall quality is now higher. The authors have made a concerted effort to improve the quality of the data analysis, which has resulted in a much stronger paper.

This manuscript is a resubmission of an earlier submission. The following is a list of the peer review reports and author responses from that submission.

Round 1

Reviewer 1 Report

Dear Authors,

the manuscript on intraspecific variations of thermal tolerance in two anuran species is of great importance, since this issue has not been extensivly studied so far. I think it provides interesting results that pose more questions to be answered. However, one thing is in my opinion, taken too much for granted, although other possibiities have not been excluded. In the behavioral part of the research, you say that toad tadpoles swim to the surface as an escape from high temperatures. However, you do not have a control group with the same tadpoles but under constant (normal) temperature to prove that the only factor responsible for this behaviour is high temperature, i.e. you do not know the "normal" behaviour of the same tadpoles. Maybe, some other things could also induce similar response (e.g. light), but you do not consider this as an option. I agree that the temperature is the most probable result, but you did not provide evidence to exclude other options.

Except for this, there are some minore things that need to be corrected that are written below:

Species names have to be in italic

Line 37- Response are determined

Figure S1- spell out HSP-abbreviation

Line 87- in additions

Line 113- spell out the author of the stages that you use

Table S1 description- Details (not detailed); n for R. uenoi is missing

Why did you use different timing for behavior experiment for the two species Please explain.

Table s1 and figure 1. n in italic and not italic. Standardize! Line 225- pattern,  not patter  Ln 244 n=? Ln 258 over the five of degree.... In discussion- not relationship but correlation Ln 286 in  light of the literature --> rephrase Ln 310 result the plasticity

Reviewer 2 Report

The authors investigated the intraspecific variation in thermal tolerance of two anuran species along a latitudinal gradient in South Korea. They measured the Critical Thermal Maximum and the behavioral responses to warming temperature from the tadpoles of Rana uenoi and Bufo sachalinensis. The authors collected an impressive amount of data for each species along this gradient. Their findings showed latitudinal variation in CTMax for R. uenoi, but no variation for B. sachalinensis. These differences between species are most likely due to variations in their thermal niches. Indeed understanding inter- and intraspecific variation in CTMax across a latitudinal gradient would help to understand how species can tolerate and adapt to climate change.

Although the authors aim to demonstrate how Critical Thermal Maximum varies across a latitudinal gradient within species, the data does not fully support this conclusion. Specifically, for R. uenoi they found the opposite of the general pattern, where species present higher CTmax close to the equator. For B. sachaliensis, no intraspecific variation across the cline was found. The authors argued that 1) different thermal niches are likely to result in variation in CTmax, and 2) daily temperature fluctuation is more important than average temperature variation across the cline. This paper would be stronger if the authors measured the microclimatic temperature variation in their habitat across the gradient. With microhabitat data, they could better link the physiological responses to the experienced environmental temperatures.

Major comments:

  1. What are the environmental temperature differences across the gradient?  

The authors used Latitude as a proxy for variation of climatic variables across the sampling sites. Latitude is merely a proxy for geographical distance. How can you differentiate between the relative roles of geographic distance and environmental variables in explaining the variation of thermal tolerance? Instead of using Latitude, maybe they could have used weather stations or microclimatic data to show the climatic cline. The environmental data they used were humidity and daylight hours. How would humidity influence the aquatic environment? How is it related to their thermal environment? ​I suggest describing better how the temperature changes across the gradient. 

  1. Why is it important to study intraspecific variation in CTmax?

Understanding how the intraspecific variation of CTmax varies across a latitudinal gradient can provide insights into how each species may respond to a human-induced climate change. For species to persist under a warming climate, they need to be able to tolerate, adapt, or migrate. In this study, the authors measured tolerance to heat stress of two anuran species and how this trait varies along a latitudinal gradient. How likely will the observed variation in CTmax help the studied species to adapt and mitigate the effect of increasing temperatures? The authors could discuss why it is important to examine intraspecific variation for a species to survive climate change (e.g. the importance of adaptive variation).

  1. How do the behavior data help to understand how the two species survive a warming temperature?

In the introduction, the authors briefly mentioned the importance of behavior to thermoregulate. They did not explicitly describe a goal and predictions on how tadpole behavior would change in different temperatures. In addition, the behavioral data does not test the ability to thermoregulate. In their experimental setup, the water temperature is homogeneous, thus, tadpoles moving up and down in the beaker do not show the ability to exploit different thermal microhabitats. I suggest removing the behavior data. 

  1. How can you use the results found in this study to extrapolate conservation efforts to other amphibians? That would be nice to see in the discussion how you could make generalizations to conserve other species with similar ecological niches. What are the other factors to consider? Larval vs. adult phases.

Minor comments:

Abstract

P1 L16-17 How did you define populations? In the methods, the authors described them as “sites.” Replace “populations” with “sites.”

Introduction

P2 L37: replace "response" with "responses"

P2 L52: replace  "increase" with "increases"

P3 L73-76: What are the characteristics of each habitat? Daily fluctuation in temperature?

P3 L73-76: How about the goals about behavior?

P3 L83: Remove "s" from "environments"

P3 L87: Remove "s" from "additions"

P3 L98: Replace "south west" with "southwest"

P3 L98: Replace  "north east" with "northeast"

Methods

P3 L94 Remove "types of"

P6 L130-132 Why daylight hours and humidity? How daylight hours and humidity were used as the independent variables? How daylight hours and humidity are expected to vary along the gradient?

P7 L158 "tapole" change to "tadpoles"

Results

P10 L257-259 Hard to read. Rewrite "the five of degrees of latitude of our study"

P11 L281-284 Hard to read: "Consequently, if R. uenoi larvae have a high risk of exposure to temperature fluctuations and cannot physiologically regulate the high metabolic demand resulting from a high DTF, then growth and development of populations inhabiting rice paddies should be negatively impacted in regions with high DTF." Rewrite

Discussion

The authors could also compare the results of CTmax for the two species with other studies (e.g., ​​Duarte et al. Global Change Biology 2012, 18, 412-421; Bennett et al. 2018. (Scientific data 2018, 5,180022).

P L340-341: How would a 1.5 degree celsius increase translate into the microhabitat of each species? Would each species be affected differently? What are the other factors to account for each species? E.g., Differences in habitat use by adults and tadpoles, genetic variability.